# Characterization of ^β^*N*-Octadecanoyl-5-hydroxytryptamide Anti-Inflammatory Effect

**DOI:** 10.3390/molecules26123709

**Published:** 2021-06-18

**Authors:** Thais Biondino Sardella Giorno, Fernanda Alves Lima, Ana Laura Macedo Brand, Camila Martins de Oliveira, Claudia Moraes Rezende, Patricia Dias Fernandes

**Affiliations:** 1Laboratório de Farmacologia da Dor e da Inflamação, Instituto de Ciências Biomédicas, Universidade Federal do Rio de Janeiro, Rio de Janeiro 21941-901, Brazil; patricia.dias@icb.ufrj.br; 2Laboratório de Análise de Aromas, Instituto de Química, Universidade Federal do Rio de Janeiro, Rio de Janeiro 21941-901, Brazil; fernanda_lima@id.uff.br (F.A.L.); alaurambrand@gmail.com (A.L.M.B.); camimartinsdeoliveira@gmail.com (C.M.d.O.); claudia.rezendeufrj@gmail.com (C.M.R.)

**Keywords:** *N*-octadecanoyl-5-hydroxytryptamide, C_18_-5HT, serotonin amides, anti-inflammatory activity, inflammation

## Abstract

Background: *N*-octadecanoyl-5-hydroxytryptamide (C_18_-5HT) is an amide that can be obtained by the coupling of serotonin and octadecanoic acid. This study aims to characterize the in vivo and in vitro anti-inflammatory activity of C_18_-5HT. Methods: A subcutaneous air pouch model (SAP) was used. The exudates were collected from SAP after carrageenan injection to assess cell migration and inflammatory mediators production. RAW 264.7 cells were used for in vitro assays. Results: C_18_-5HT significantly inhibited leukocyte migration into the SAP as well as nitric oxide (NO) and cytokines production and protein extravasation. We also observed an reduction in some cytokines and an increase in IL-10 production. Assays conducted with RAW 264.7 cells indicated that C_18_-5HT inhibited NO and cytokine produced. Conclusions: Taken together, our data suggest that C_18_-5HT presents a significant effect in different cell types (leukocytes collected from exudate, mainly polumorphonuclear leukocytes and cell culture macrophages) and is a promising compound for further studies for the development of a new anti-inflammatory drug.

## 1. Introduction

Inflammation is a dynamic response to tissue injury and pathogen invasion. Macrophages are the first cells to recognize the potential threat, followed by mast cells, leukocytes, and neutrophils. These cells release inflammatory mediators that amplify the immune response, such as cytokines, nitric oxide (NO), chemokines, vasoactive amines, leukotrienes, and prostaglandins (PGs), and they promote fever, redness, edema, and pain [1]. Cytokines released by cells(e.g., tumor necrosis factor-α (TNF-α) and interleukins (ILs) bind to G protein-coupled receptors (GPCRs) to repair local damage, inducing the expression of selectins and integrins. The chemokines produced by resident cells recruit, through their receptors, circulating neutrophils [2]. These neutrophils transmigrate through the endothelium, reach the inflamed tissue, degranulate, generate reactive oxygen species (ROS), and produce other mediators such as NO and PGs, amplifying the inflammatory response [2].

Fatty acid conjugates with ethanolamine, amino acids, or mono-amine neurotransmitters occur in nature. They are subdivided into *N*-acyl ethanolamines (NAE) and *N*-acyl amines (NAA), each with different biological activities [3,4]. The most famous representative of NAA is anandamide (AEA, *N*-arachidonoyl ethanolamine), which is an endocannabinoid known for its pleiotropic effect varying from homeostasis to immune functions [5]. It acts as an agonist in cannabinoid receptors 1 (CB1) and 2 (CB2), exerting different activities in both the brain and the periphery [5]. Docosahexaenoyl ethanolamine (DHEA), a structural analogue of AEA, exhibits anti-inflammatory properties, modulating the activity of cyclooxygenase-2 (COX-2), which is the main enzyme in the synthesis of pro-inflammatory PGs. In contrast, there is not much information available in the literature on fatty acid conjugates with 5-hydroxytryptamide, which are known as *N*-acyl serotonins.

The first investigated *N*-acyl serotonin was arachidonoyl serotonin (AA-5-HT), which is a derivative of the conjugation of arachidonic acid with serotonin. AA-5HT inhibits fatty acid amide hydrolases (FAAH) and is an agonist of type 1 transient potential receptor (TRVP1). It has potential anxiolytic, antiallergic, and analgesic properties [6]. DHA-5HT, a conjugate of docosahexaenoic acid and serotonin, presents potent immunomodulatory properties by attenuating the IL-23-IL-17 signaling in macrophages stimulated with lipopolysaccharides (LPS) [7].

Recently, we demonstrated that *N*-octadecanoyl serotonin (C_18_-5HT) has antinociceptive action in the formalin-, capsaicin-, and glutamate-induced licking nociception and hot plate models. This effect is mediated, in part, by the opioid, serotoninergic, and cannabinoid systems. C_18_-5HT also showed an anti-hyperalgesic effect [8]. Our group also demonstrated that ^β^*N*-arachinoyl-5-hydroxytryptamide (C_20_-5HT) and ^β^*N*-behenoyl-5-hydroxytryptamide (C_22_-5HT) presented a significant antinociceptive effect [9].

Thus, this study aims to investigate the effect of serotonin derivative C_18_-5HT on in vivo and in vitro models of inflammation.

## 2. Results

### 2.1. C_18_-5HT Synthesis

C_18_-5HT was synthesized using an adapted protocol by Reddy and Swamy [10]. The NRM spectra, melting point, and exact mass of the pure product are per the literature [10,11].

### 2.2. C_18_-5HT Does Not Affect Cell Viability and Inhibit Nitric Oxide and Cytokines Production

Figure 1A shows that concentrations C_18_-5HT ranging from 0.01 to 1 µM did not alter the cell viability after 24 or 48 h of incubation. However, it could be noted that 3 µM of C_18_-5HT significantly reduced cell viability (in 25%). Hence, the maximum concentration of 1 µM was used in the following assays. Next, we investigated the effect of C_18_-5HT on NO production following LPS (1 µg/mL)activation. LPS activation led to a 3.84-fold increase in NO production (13.3 ± 3.2 µM to 50 ± 1.4 µM in vehicle-treated and LPS-activated groups, respectively). When LPS-activated cells were further incubated with C_18_-5HT (0.1, 0.3, or 1 µM), a significant and dose-dependent inhibition of NO production was observed (4%, 20%, and 38%, respectively) (Figure 1B).

We also evaluated a possible inhibitory effect of C_18_-5HT in cytokine production by RAW 264.7 cells. All three concentrations of the compound significantly reduced IL-1β production in a dose-dependent way.On the other hand, there were lower levels of TNF-α and IL-6 and higher levels of IL-10 only after incubation with 1 µM of C_18_-5HT cells (Figure 2).

### 2.3. C_18_-5HT Also Presents an Effect When Orally Administered to Mice

Continuing the studies with C_18_-5HT, we decided to assess whether it had an effect when administered orally to mice. In this model, twenty-four hours after carrageenan injection into the subcutaneous air pouch (SAP), animals from carrageenan-injected group showed a 20.86-fold increase in cell migration (3.5 ± 1.4 × 10^6^ cells/mL and 73 ± 10 × 10^6^ cells/mL, to saline-and carrageenan-treated groups, respectively). Contrarily, the treatment of mice with dexamethasone (2.5 mg/kg, i.p.) 1 h prior to carrageenan injection into the SAP reduced the cell migration to the cavity by 64.3%. Crescent doses of C_18_-5HT (0.1, 1 or 10 mg/kg, p.o.) also reduced cell migration to the SAP induced by carrageenan by 32.9%, 47.9%, and 49.3%, respectively (Figure 3).

To understand if the effect of C_18_-5HT on cell migration occurred due to an inhibition of the transmigration of cells from the blood to the cavity or because of a decrease in the number of cells produced by the bone marrow, we counted the number of leukocytes in the blood and bone marrow collected from animals treated with C_18_-5HT (10 mg/kg). The data obtained showed that C_18_-5HT did not demonstrate any hematological alteration or myelotoxicity, since there were no alterations in the number of cells in the blood and the bone marrow (data not shown).

### 2.4. Pretreatment of Mice with C_18_-5HT Reduces Protein Extravasation, Nitric Oxide, and Cytokines Production

As expected, carrageenan administration into the SAP led to a 10- and 9.5-fold increase in protein extravasation and NO production, respectively. In the mice pretreated with dexamethasone (2.5 mg/kg, i.p.), the reduction observed was 45.9% and 68% to both parameters. The oral administration of C_18_-5HT 1 h previous to carrageenan injection in the SAP caused a dose-dependent reduction in the amount of protein extravasated to the exudate, with a reduction in 15.6%, 27.7%, and 35.2%, to 0.1, 1, and 10 mg/kg, respectively. The same doses also reduced the levels of NO measured in the fluid (51.2%, 64%, and 75%, respectively) (Figure 4).

We subsequently measured the levels of some cytokines (i.e., TNF-α, IL-1β, IL-10, and IFN-γ) in the exudates collected in SAP after the treatment of mice. Figure 5 shows that carrageenan causes a significant increase in levels of all four cytokines 24 h after its injection in the SAP. The pretreatment of mice with dexamethasone (2.5 mg/kg, i.p.) reduced by more than 50% the levels of TNF-α, IL-1β, and IFN-γ (599.7 ± 8.9 pg/mL in the carrageenan group versus 257.8 ± 21.8 pg/mL, 282.9 ± 62.6 pg/mL, and 325.4 ± 33.1 pg/mL of each cytokine, respectively), and increased production of IL-10 by almost 100% (2525 ± 215.7 pg/mL in the carrageenan group versus 5051 ± 431.4 pg/mL in the dexamethasone-treated group). Although all doses of C_18_-5HT (0.1, 1, or 10 mg/kg, p.o.) significantly reduced TNF-α levels (55.98 ± 8.9 pg/mL in the saline-injected group versus 599.7 ± 83.15 pg/mL in the carrageenan-treated group, and 396.1 ± 40.7 pg/mL, 317.1 ± 58 pg/mL, and 271.5 ± 36.15 pg/mL to 0.1, 1, and 10 mg/kg doses, respectively), only the two higher doses inhibited IL-1β production (768.5 ± 149.4 pg/mL in the saline-treated group versus 1584 ± 350.7 pg/mL in the carrageenan-treated group and 1323 ± 192.3 pg/mL, 733.8 ± 69.1 pg/mL, and 622.9 ± 53.18 pg/mL to 0.1, 1, and 10 mg/kg doses, respectively). Regarding the cytokine IFN-γ, only the dose of 10 mg/kg demonstrated a significant effect (62.5 ± 25.5 pg/mL in the saline-treated group versus 1302 ±132.6 pg/mL in the carrageenan-treated group and 1212 ± 185.6 pg/mL, 1273 ± 151.2 pg/mL, and 808 ± 128.2 pg/mL to 0.1, 1, and 10 mg/kg doses, respectively). The 10 mg/kg dose also increased IL-10 production and accumulation in the exudate (905.6 ± 59.2 pg/mL in the saline-treated group versus2525 ± 215.7 pg/mL in the carrageenan-treated group and 4214 ± 453.8 pg/mL in 10 mg/kg) (Figure 5).

### 2.5. C_18_-5HT Reduces Reactive Oxygen Species Production by Leukocytes in an Ex Vivo Assay

We also decided to evaluate a possible antioxidant effect of C_18_-5HT using an ex vivo assay with leukocytes collected from the exudate. Figure 6 shows that all concentrations of C_18_-5HT (0.1, 0.3, and 1 µM) almost completely abolished reactive oxygen species (ROS) produced by PMA-activated cells. Values obtained in the C_18_-5HT-treated groups were similar to those obtained in the non-PMA stimulated group. These results corroborate the anti-inflammatory effects observed in the previous results obtained in the carrageenan-induced cell migration and in vitro assays.

## 3. Discussion

Previous studies by our group showed that C_18_-5HT has an antinociceptive effect in peripheral and central nociception models [8]. We also demonstrated that the serotonin amide presented affects the second phase of the formalin-induced licking response, which is a well-established model for the screening of possible antinociceptive and anti-inflammatory compounds. In the second phase of this model, there is a release of inflammatory mediators that act together in nociceptors and local receptors [12,13,14]. The involvement of serotonin and bradykinin in both phases has also been described [15].Thus, we decided to evaluate a possible anti-inflammatory effect of C_18_-5HT.

Acute inflammation is an innate immune system response initiated by a wide range of injuries (physical, chemical, or biological)that involves a cascade of biochemical and cellular events, which include fluid extravasation, enzymatic activation, cell migration, mediator release, receptor sensitization, activation, lysis, and tissue repair [16]. One of these events is the cellular transmigration from the circulation to the injured site. The migration of leukocytes from the circulation to the site of inflammation is mediated by molecular interactions between neutrophils, endothelial cells, and components of the extracellular matrix [17].The neutrophil is the main type of leukocyte involved in the innate mechanisms of defense. Its recognition is critical for the host’s response against the invasion of tissues by microorganisms [18]. Neutrophils are attracted to sites of inflammation by various stimuli, such as microorganism products (LPS) and chemotactic factors released by resident cells, such as cytokines, chemokines, and eicosanoids [19,20,21]. In addition, neutrophils also release cytokines and chemokines that promote their self-activation, as well as the recruitment of other cells of the immune system [22].

The carrageenan-induced cell migration model is widely used to screen drugs with potential anti-inflammatory action. The injection of carrageenan into the subcutaneous air pouch induces an inflammatory process with characteristics and development similar to that observed in rheumatoid arthritis, with polymorphonuclear infiltration and the release of pro-inflammatory mediators [23].In the initial phase (2–4 hs) after the carrageenan injection, an intense process of leukocyte rolling and cell adhesion occurs. Six hours after the injection, there is a large influx of neutrophils and the release of chemotactic agents at the inflammatory site [24]. After 24 h cellular influx, exudate formation and the production of inflammatory mediators (i.e., NO, PGs, leukotrienes, cytokines)are augmented [25,26,27,28,29].

Our results showed that C_18_-5HT reduced leukocyte migration into the cavity. To explain this effect, we hypothesize that C_18_-5HT could inhibit the production and/or release of pro-inflammatory substances produced by leukocytes located in the pouch. Thus, we investigated if C_18_-5HT could also reduce the production of some inflammatory mediators, such as NO and the cytokines TNF-α, IL-1β, and INF-γ.

The fact that C_18_-5HT reduced the levels of TNF-α, IL-1β, and INF-γ produced in the exudate could be correlated with a direct reduction in leukocyte migration or an effect in leukocytes, inducing an inhibitory effect in the cells. These results are in accordance with those reported by other studies. Rau and collaborators observed that the immunosuppressant tacrolimus inhibited the infiltration of neutrophils in rats via reduction of the expression of the mRNA for TNF-α and IL-1β in a model of acute pancreatitis induced by taurocholate [30]. We also observed that C_18_-5HT increased the levels of IL-10,which is an immunomodulatory cytokine that, together with the pro-inflammatory cytokines, participates in the expression of adhesion molecules. These molecules are important for the firm adhesion and diapedesis of leukocytes. Additionally, they form a complex network of communication that results in effects that are not determined by the action of a single cytokine but by the interrelationship between several cytokines [31]. Thus, taken together, these data suggest that C_18_-5HT may act on one or more cellular events by inhibiting the production of mediators that stimulate the expression of adhesion molecules (such as pro-inflammatory cytokines) or pro-inflammatory transcription factors such as NF-κB.

These inflammatory mediators can also cause increased vascular permeability and consequent protein extravasation [32,33]. Pretreatment of mice with C_18_-5HT resulted in a corresponding reduction in protein extravasation, thus confirming and suggesting the correlation between the inflammatory mediators and plasma and protein leakage. This effect highlights the action of this compound on inflammatory mediators involved in increasing vascular permeability, such as cytokines and NO.

The effects of NO along the inflammatory response are complex and involve a series of events. Our results showed that oral pretreatment with C_18_-5HT significantly inhibited NO production.This decrease may be a consequence of reduced cell transmigration to the cavity and/or inhibition of the effects of cytokines, eicosanoids, kinins, and glutamate or even alterations in the expression and/or activity of inducible nitric oxide synthase (NOS) enzymes [34,35]. Our results support the first hypothesis, since the reduction in NO production was proportional to the reduction in the number of leukocytes that migrate to the cavity.Farsky et al. observed that systemic pretreatment with L-nitroarginine methyl ester (L-NAME), a NOS inhibitor, inhibits the influx of leukocytes into the air pouch cavity in mice injected with *Bothrops jararaca* venom [36]. Tunon et al. demonstrated that tacrolimus inhibits the expression of iNOS mRNA in hepatocytes from rats incubated with LPS [37]. Finally, Sakaguchi et al. (2006) observed that intraperitoneal administration of L-NAME in rats reduced exudate formation and albumin levels in carrageenan-induced pleurisy [38]. These studies corroborate our results, as they highlight the influence of NO production on neutrophil infiltration.

Higher oxidative stress imbalance activates redox-sensitive factors, such as NF-κB, induces inflammatory cytokines, and promotes inflammation [39,40,41].Thus, inflammatory cytokines activate sequential events, increase oxidative stress, and contribute to the progression of the disease by the intensification of the inflammatory response [39].TNF-α synthesis is inhibited by IL-10, which is an immunomodulatory cytokine that reduces ROS formation and NO production. TNF-α also increases the levels of ROS, which is prevented by the action of IL-10 [39]. C_18_-5HT reduced ROS formation in PMA-stimulated leukocytes. It also increased IL-10 and reduced TNF-α and NO levels in the collected exudate. Together, these results suggest that C_18_-5HT has antioxidant and anti-inflammatory effects.

Macrophages are phagocytic cells of the innate immune system that are responsible for recognizing an infection, presenting antigens, and also for the production of inflammatory mediators, the removal of apoptotic cells, and the output of growth factors. The subtype M1 is activated by pro-inflammatory cytokines or a microorganism’s molecules (such as LPS), and it has killing capacity by the production of ROS, NO, and inflammatory cytokines [42].

Non-steroidal anti-inflammatory drugs and some anesthetics inhibit the production of pro-inflammatory cytokines and chemokines, including TNF-α, IL-1β, IL-6, and IL-8,as well as NO synthesis [43]. Some drugs also modulate the production of immunomodulatory cytokines, such as IL-10, which is a cytokine capable of modulating negatively and indirectly the synthesis of these inflammatory mediators and surface molecules [44].

As C_18_-5HT showed an in vivo anti-inflammatory effect, we decided to investigate whether it would also have some in vitroactivity, using the LPS-stimulated RAW 264.7 cell line to evaluate the immunomodulatory effect of C_18_-5HT. The data obtained showed that C_18_-5HT did not present a cytotoxic effect. We also demonstrated that the serotonin amide did reduce NO production by LPS-activated RAW 264.7 cells.Our results show that the anti-inflammatory effect of C_18_-5HT may be related to the inhibition of the synthesis of TNF-α and IL-1β by LPS-stimulated macrophages. It also positively modulated the release of IL-10. These results suggest that the anti-inflammatory effect of C_18_-5HT may be IL-10 dependent, and it probably has a direct effect on the transcription and/or translation of these inflammatory mediators.

The anti-inflammatory activity of C_18_-5HT, a conjugate of octadecanoic acid with serotonin, is related to the fatty acid side chain (18 carbons). PA-5HT, a conjugate of palmitic acid (16:0) and serotonin, has an anti-inflammatory effect in RAW 264.7 cells and anti-allergic actions in RBL-2H3 cells [45,46]. *N*-acyl serotonin derivatives with longer and unsaturated alkyl chains, such as the arachidonic acid (AA-5-HT (20: 4)), eicosapentaenoic acid (EPA-5-HT (20: 5)), and docosahexaenoic acid (DHA-5-HT (22: 6)) conjugates, have greater ability to suppress IL-17 production in macrophages [3]. Thus, the anti-inflammatory potency of the serotonin amides and their analogues is directly related to the fatty acid moiety of the molecule.

## 4. Materials and Methods

### 4.1. C_18_-5HT Synthesis

C_18_-5HT was synthesized using a condensation reaction between stearic acid and serotonin, which was adapted from Reddy and Swamy [10]. Stearic acid (1.1 eq), DMAP (1.0 eq), serotonin hydrochloride (1.0 eq.), and EDC.HCl (1.0 eq.) were dissolved in DMF. The mixture was stirred at room temperature for 24 h and monitored via thin-layer chromatography (TLC). The crude product was purified on a silica gel column by using increasing concentrations of ethyl acetate in dichloromethane (DCM). C_18_-5HT was characterized by 1H- and 13C-NMR spectroscopy, melting point determination, and high-resolution mass spectrometry. NMR spectra were obtained using a Varian VNMRS 500 MHz spectrometer. The product exact mass was analyzed in a Dionex UltiMate 3000 liquid chromatography coupled to a hybrid Quadrupole-Orbitrap high-resolution mass spectrometer (Thermo Q-Exactive Plus, Thermo Fisher Scientific, Waltham, MA, USA) equipped with an electrospray ionization (ESI) source. A Syncronis C-18 column (50 mm × 2.1 mm × 1.7 µm) was used with gradient elution mode with water (mobile phase A) and methanol (mobile phase B) both with 0.1% formic acid and 5 mM ammonium formate as follows: 0–4 min 5–30% B, 4.1–10.0 min 30–50% B, 10–14 min 50–98% B, 14–17 min 98% B, and 17.1–22 min 5% B. The column temperature was set to 40 °C, the solvent flow rate was 0.350 mL/min, and the injection volume was 8 µL. Mass spectrometry conditions were spray voltage 3.9 kV (ESI+); ion transfer capillary 300 °C; sheath and auxiliary gases 50 and 15 arbitrary units, respectively; and normalized collision energy (NCE) of 22 for positive mode. Data were acquired by a parallel reaction monitoring (PRM) experiment at a resolution of 35.000, producing an inclusion list with C_18_-5HT theoretical exact mass.

*^β^**N-Stearoyl-5-Hydroxytryptamide (C_18_-5HT)Data*, White solid; 77.8% yield (121.06 mg); mp 118.5–119.0 °C; 1H NMR (500 MHz, DMSO-d6): δ 0.84 (3H, t, J = 6.9 Hz, CH3), 1.23 (28H, s, H30-H16H), 1.48 (2H, m, H2,), 2.04 (2H, t, J = 6.5 Hz, H1,), 2.70 (2H, t, J = 6.8 Hz, H8), 3.27 (2H, m, H9), 6.58 (1H, d, J = 8.2 Hz, H6), 6.82 (1H, s, H4), 7.01 (1H, s, H2), 7.11 (1H, d, J = 8.2 Hz, H7), 7.86 (1H, m, NH-CO), 8.59 (1H, s, OH), 10.47 (1H, s, NH) ppm; 13C NMR (500 MHz, DMSO-d6): δ 13.97 (CH3), 22.14 (C31–C18C), 25.35 (C2,), 25.47 (C8), 28.76 (C3′–C18′), 28.88 (C3′–C18′), 29.01 (C3′–C18′), 29.06 (C3′–C18′), 29.09 (C3′–C18′), 31.34 (C3′–C18′), 35.52 (C1′), 39.52 (C9), 102.23 (C4), 110.92 (C3), 111.24 (C6), 111.62 (C7), 122.98 (C2), 127.92 (C3a), 130.82 (C7a), 150.18 (C5), 171.98 (NH-CO) ppm. HRMS (ESI) for C_28_H_46_N_2_O_2_ [M+H]+ Exact Mass: 443.36320, found 443.36362.

### 4.2. Materials and Reagents

Carrageenan (type IV), dexamethasone, lipopolysaccharide (LPS), Roswell Park Memorial Institute (RPMI) 1640 medium, fetal bovine serum (FBS), *N*-(3-dimethylaminopropyl)-N′-ethylcarbodiimide hydrochloride 98% (EDC.HCl), 4-dimethylaminopyridine (DMAP), serotonin hydrochloride, and stearic acid were acquired from Sigma (St. Louis, MO, USA). The RAW 264.7 murine macrophage cell line was purchased from American Type Culture Collection (ATTC TIB-71). ELISA antibodies and Kit Pierce BCA™ Protein Assay (Thermo Scientific, NY, EUA) were obtained from BD Biosciences (New York, NY, USA) and Thermo Fisher Scientific, Inc (Waltham, MA, USA), respectively. *N,N*-Dimethylmethanamide 99.8% (DMF), dichloromethane 99.8% (DCM), and ethyl acetate 99.5% were purchased from Merck Inc. (Sao Paulo, Brazil).

### 4.3. Drugs and C_18_-5HT Administration

C_18_-5HT was dissolved in dimethylsulfoxide (DMSO) to prepare a 100 mg/mL stock solution. On the day of the experiments, this solution was diluted in tween 80 and administered by oral gavage to the animals at the doses of 0.1, 1, or 10 mg/kg in a final volume of 0.1 mL. The vehicle group was composed of tween 80 (negative control group) and the positive control group was composed of dexamethasone (2.5 mg/kg, i.p.). All drugs were diluted immediately before use.

### 4.4. Cell Culture

RAW 264.7 murine macrophages (ATCC TIB-71) were cultured in RPMI 1640 medium supplemented with 10% fetal bovine serum and glutamine (2 mM). Cells were maintained in a humidified atmosphere containing 5% CO_2_ at 37 °C, and the medium was changed every 2 or 3 days. Cells between the 4^th^ and 6^th^ passages were used for different pellet preparations.

### 4.5. Cell Viability

The cell viability was evaluated by MTT (3-(4,5-dimethyl-thiazol-2-yl)-2,5-diphenyltetrazolium bromide) assay. The mitochondrial-dependent reduction of MTT to formazan was used to measure cell respiration as an indicator of cell viability [47]. When the cells formed a confluent monolayer, they were scrapped, centrifuged, plated in a 96-well plate at a density of 2.5 × 10^4^ cells/mL, and incubated with different concentrations of C_18_-5HT. After 24 and 48 h, 20 µL of MTT (5 µg/mL) was added to each well, and the plates were maintained at 37 °C for 4 h. After the incubation period, the supernatant was discarded, and DMSO (100 µL) was added to dissolve MTT formazan crystals. The absorbance was read in a FlexStation reader at 570 nm (Molecular Devices, San Jose, CA, USA).

### 4.6. Animals

The male Swiss Webster mice (20–25 g, 8–10 weeks) used in all experiments were donated by the Instituto Vital Brazil (Niterói, Rio de Janeiro, Brazil). The animals were kept in a controlled temperature environment (22 ± 2 °C, 60% to 80% humidity) and light–dark cycles of 12 h. Food and water were provided ad libitum, but the feed was withdrawn 1 h before oral administration of substances. The protocol for this study was approved by the National Council for the Control of Animal Experimentation (CONCEA), Biomedical Science Institute/UFRJ, and Ethical Committee for Animal Research, under the number 35/19.

### 4.7. Carrageenan-Induced Cellular Migration into the Subcutaneous Air Pouch (SAP)

Carrageenan-induced cell migration was performed according to Raymundo et al. [48]. The mice’s backs were injected subcutaneously with 10 mL of sterile air. Three days later, a new injection with 8 mL of sterile air was made to guarantee the maintenance of the cavity in the animal’s back. On the sixth day, the animals were treated orally with the vehicle, C_18_-5HT (0.1, 1,or10 mg/kg) or intraperitoneally with dexamethasone (2.5 mg/kg) 60 min before receiving 1 mL of a carrageenan solution (1%). After 24 h of carrageenan injection, the animals were euthanized with chloral hydrate (1%, i.p.), and the cavity was flushed out with 1 mL of PBS.Exudates were collected and centrifuged at 1200 rpm for 10 min at 4 °C, and samples were stored at −20 °C until the quantifications. The total leucocytes cells counts were determined in the exudates using a pocH-100iV Diff (Sysmex) hematology analyzer.

### 4.8. Nitrate, Nitrite, Protein, and Cytokines Measurement

To evaluate the nitrate accumulated in SAP, exudates were analyzed according to the method described by Bartholomew [49] and adapted by Raymundo et al. [48], which was followed by the measurement of nitrite according to the Griess reaction [50].

Protein quantification was performed by the BCA method using the protein dosage kit BCA^TM^ (Thermo Fisher Scientific Inc., Whaltman, MA, USA). At the time of the dosage, reagents A (sodium carbonate, sodium bicarbonate, bicinchoninic acid, and sodium tartrate in 0.1 M of sodium hydroxide and B (4% copper sulfate) were mixed in a ratio of 5:1. A volume of 5 µL of the sample was incubated with 195 µL of the BCA reagent for 30 min at 37 °C, and the absorbance was measured in a microplate reader at 562 nm.

Supernatants from the exudates collected from the SAP were used to measure the levels of the cytokines by enzyme-linked immunosorbent assay (ELISA) using the protocol supplied by the manufacturer (B&D, Franklin Lakes, NJ, USA).

### 4.9. Ex Vivo Reactive Oxygen Species (ROS) Measurement

Leukocytes collected from SAP were placed in tubes (1 × 10^6^ cells/mL) and incubated for 1 h (37 °C and 5% CO_2_). After incubation with C_18_-5HT (0.1, 0.3, and 1 µM) for 30 min and stimulated with phorbol myristate acetate (PMA, 10 nM) for 45 min, DCF-DA (2′-7′diclorodihidrofluorescein diacetate, 2 mM) was added, and the cells were incubated for 30 min (37 °C and 5% CO_2_) [51]. The emitted fluorescence was captured in the FL-1 channel flow cytometer (BD Accuri™, B&D, Franklin Lakes, NJ, USA) and was expressed as DCF-DA fluorescence.

### 4.10. Statistical Analysis

Results are expressed as the mean ± standard deviation (S.D.). Statistical significance was calculated by analysis of variance (ANOVA) followed by Newman’s post-test. *p* values less than 0.05 (* *p* < 0.05) were considered significant.

In vivo, experimental groups were composed of 6–8 animals. For in vitro assays, each protocol was repeated at least 3 times, and each concentration tested was done in triplicate (n = 3).

## 5. Conclusions

We showed that C_18_-5HT has potent in vivo and in vitro anti-inflammatory effects, suggesting that this compound may be a prototype for the development of a new compound for further tests.

## Figures and Tables

**Figure 1 molecules-26-03709-f001:**
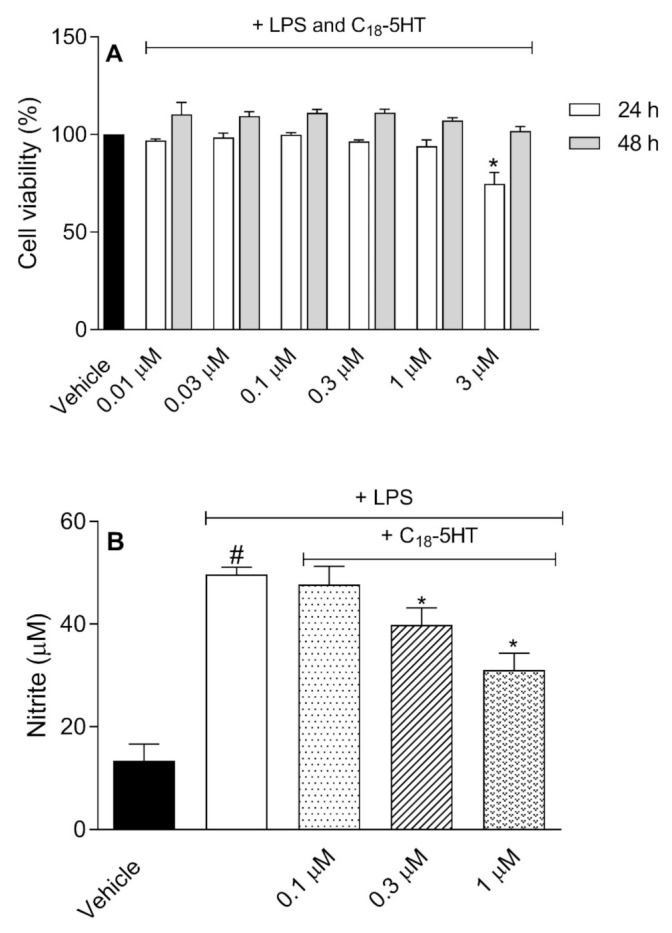
Effect of C_18_-5HT on cell viability (**A**) and NO production (**B**) in a RAW 264.7 cell line. In A, cells were incubated with vehicle or C_18_-5HT (0.01–3 µM) and further incubated with lipopoly, (1 μg/mL) for 24 or 48 h. Cell viability was measured with an MTT assay. In B, cells were incubated with vehicle or C_18_-5HT (0.1–1 µM) and further incubated with LPS (1 μg/mL). After 24 h of incubation, nitrite concentration in the supernatant was measured with Griess reagent. The results are expressed as mean ± SD (n = 3) from three independent experiments. In (**A**), * *p* < 0.05 when compared with the vehicle-treated group. In (**B**), * *p* < 0.05 when comparing the C_18_-5HT-treated group with LPS-treated group or # *p* < 0.05 when comparing the LPS-treated group with the vehicle-treated group.

**Figure 2 molecules-26-03709-f002:**
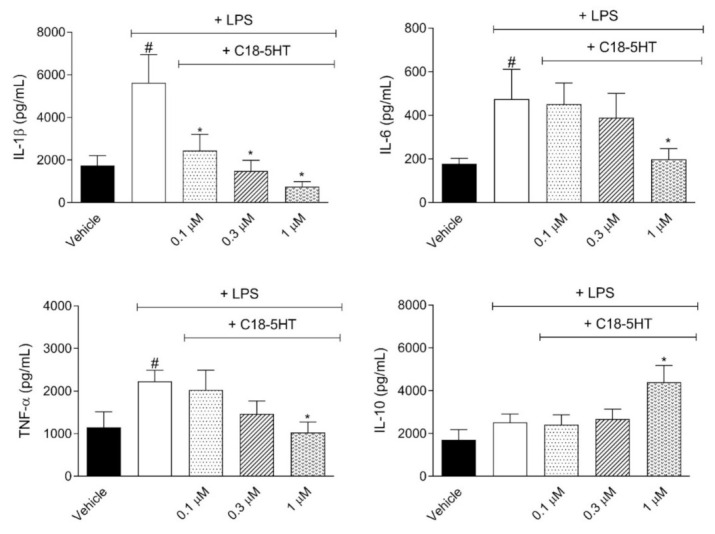
Effect of C_18_-5HT on IL-1β, IL-6, TNF-α,and IL-10 produced by LPS-stimulated RAW 264.7. Cells were treated with C_18_-5HT (0.1, 0.3, or 1 µM) for 30 min and then stimulated with LPS (1 μg/mL) for 24 h. Cytokines were determined by the ELISA method. The results are expressed as mean ± SD (n = 3) from three independent experiments. * *p* < 0.05 when comparing the C_18_-5HT-treated group with the LPS-treated group or # *p* < 0.05 when comparing the LPS-treated group with the vehicle-treated group.

**Figure 3 molecules-26-03709-f003:**
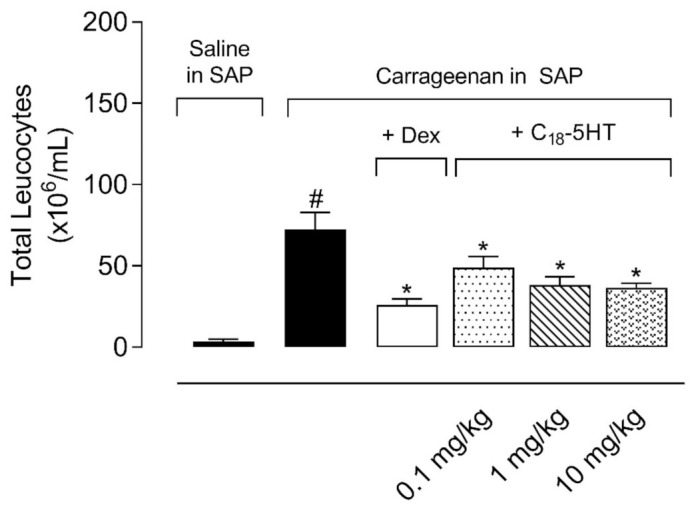
C_18_-5HT reduced cell migration induced by carrageenan into the subcutaneous air pouch (SAP). The animals were pretreated with vehicle (tween 80, p.o.), C_18_-5HT (0.1, 1, or 10 mg/kg, p.o.), or dexamethasone (Dex, 2.5 mg/kg, i.p.) 60 min before carrageenan (1%, *w*/*v*) injection into SAP. Results are expressed as mean ± S.D. (n = 6–8). The statistical significance was calculated by ANOVA followed Newman’s post-test. # *p* < 0.05 when comparing the vehicle-treated group that received carrageenan in SAP with the vehicle-treated group that received saline in SAP and * *p* < 0.05 when comparing the dexamethasone or C_18_-5HT pretreated groups that received carrageenan in SAP with the vehicle-treated group that received carrageenan in SAP.

**Figure 4 molecules-26-03709-f004:**
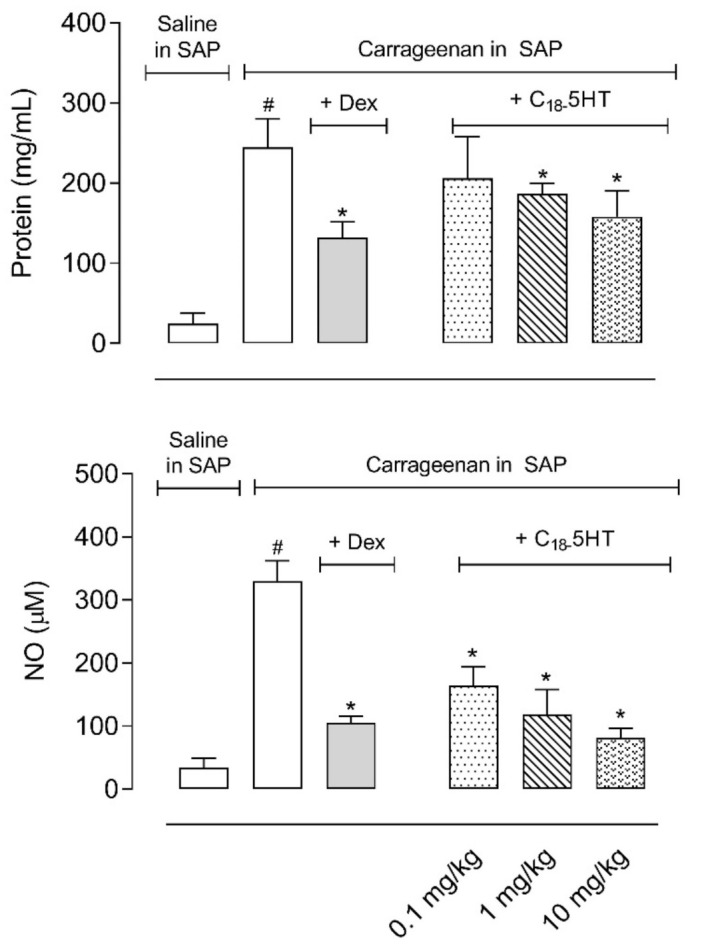
C_18_-5HT reduces protein extravasation and NO production into the subcutaneous air pouch (SAP). The animals were pretreated with vehicle (tween 80, p.o.), C_18_-5HT (0.1, 1, or 10 mg/kg, p.o.), or dexamethasone (Dex, 2.5 mg/kg, i.p.) 60 min before carrageenan (1%) injection into SAP. Results are expressed as mean ± S.D. (n = 6–8). The statistical significance was calculated by ANOVA followed by Newman’s post-test. # *p* < 0.05 when comparing the vehicle-treated group that received carrageenan in SAP with the vehicle-treated group that received saline in SAP and * *p* < 0.05 when comparing the dexamethasone or C_18_-5HT pretreated groups that received carrageenan in SAP with the vehicle-treated group that received carrageenan in SAP.

**Figure 5 molecules-26-03709-f005:**
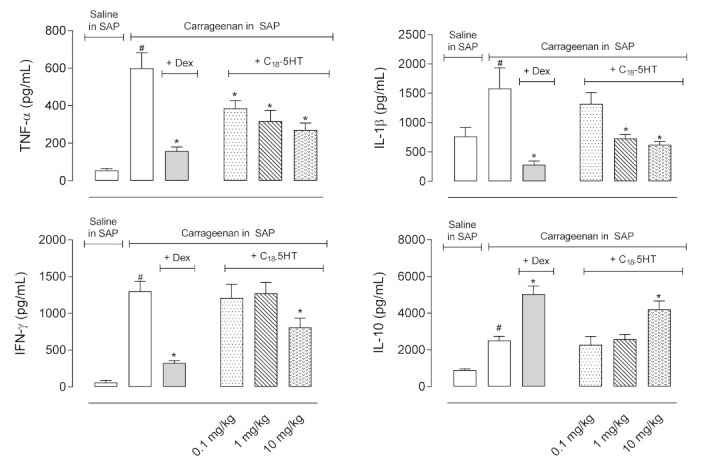
C_18_-5HT reduces cytokine production in the subcutaneous air pouch (SAP). The animals were pretreated with vehicle (tween 80, p.o.), C_18_-5HT (0.1, 1 or 10 mg/kg, p.o.), or dexamethasone (Dex, 2.5 mg/kg, i.p.) 60 min before carrageenan (1%) injection into SAP. Results are expressed as mean ± S.D. (n = 6–8). The statistical significance was calculated by ANOVA followed Newman’s post-test. # *p* < 0.05 when comparing the vehicle-treated group that received carrageenan in SAP with the vehicle-treated group that received saline in SAP and * *p* < 0.05 when comparing the dexamethasone or C_18_-5HT pretreated groups that received carrageenan in SAP with the vehicle-treated group that received carrageenan in SAP.

**Figure 6 molecules-26-03709-f006:**
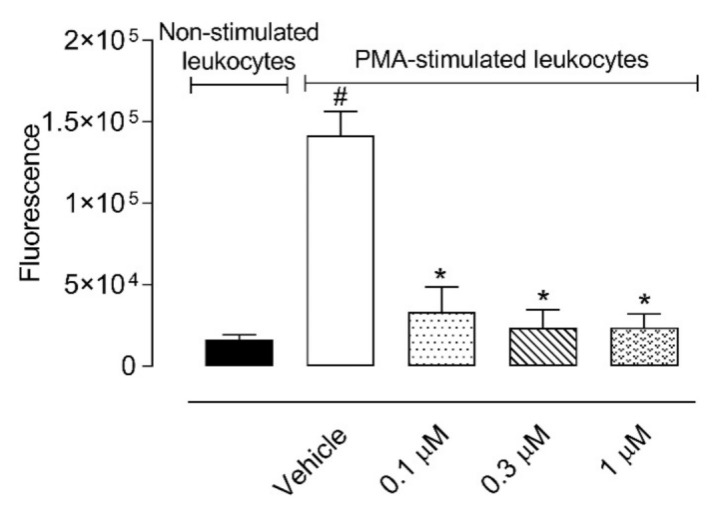
C_18_-5HT presented an antioxidant effect reducing reactive oxygen species (ROS) production. Leukocytes obtained after carrageenan injection into the subcutaneous air pouch were used ex vivo. Cells were activated with PMA and further incubated with vehicle or C_18_-5HT (0.1, 0.3, or 1 µM). Fluorescence intensity was measured using DCF-DA. Results are expressed as mean ± S.D. of fluorescence intensity. The statistical significance was calculated by ANOVA followed Newman’s post-test. # *p* < 0.01 when comparing PMA-stimulated leukocytes incubated with vehicle with non-stimulated leukocytes and * *p* < 0.05 when comparing PMA-stimulated leukocytes incubated with C_18_-5HT with PMA-stimulated leukocytes.

## Data Availability

Not applicable.

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
