# Peer review of "Characterization of βN-Octadecanoyl-5-hydroxytryptamide Anti-Inflammatory Effect"

_molecules, 2021, doi:10.3390/molecules26123709_

Round 1

Reviewer 1 Report

In the manuscript, “Pre-clinical evaluation of βN-octadecanoyl-5-hydroxytryptamide: a new option as future anti-inflammatory compound?” the authors have tried to examine N-octadecanoyl-5-hydroxytryptamide (C18-5HT) as an anti-inflammatory drug. The article is nicely written, and experimental execution is also acceptable. I found the study to be interesting and suggest acceptance of the manuscript after some necessary minor revision. The comments are as follows:

  • Authors are advised to reconsider the title as it should be more realistic with the findings of the manuscript.
  • Can the authors explain why toxicity was seen after 24h of 3 µM (C18-5HT) treatment, not after 48h?

Author Response

In the manuscript, “Pre-clinical evaluation of βN-octadecanoyl-5-hydroxytryptamide: a new option as future anti-inflammatory compound?” the authors have tried to examine N-octadecanoyl-5-hydroxytryptamide (C18-5HT) as an anti-inflammatory drug. The article is nicely written, and experimental execution is also acceptable. I found the study to be interesting and suggest acceptance of the manuscript after some necessary minor revision. The comments are as follows:

We really appreciate the comment.

-Authors are advised to reconsider the title as it should be more realistic with the findings of the manuscript.

We changed the title to “Characterization of βN-octadecanoyl-5-hydroxytryptamide: a new option as future anti-inflammatory compound?”

-Can the authors explain why toxicity was seen after 24h of 3 µM (C18-5HT) treatment, not after 48h?

We really don’t have an explanation to this observation. One possibility could be that, during the first 24 h incubation, the substance is still “intact” in medium and after 24 hours it could degrade due to the presence of any esterase presente in fetal bovine serum that is added to culture medium. However, in the past we did some assays to evaluate a possible degradation of the substance and in conditions without cell we did not observe any degratation.

Reviewer 2 Report

The authors analyzed the anti-inflammatory activity of C18-5HT, and found that C18-5HT inhibits the leukocyte migration into the SAP, the production of NO and cytokines, and the protein extravasation. The study is interesting and potentially important for the development of new anti-inflammatory drugs. To step up the quality of the manuscript, the reviewer suggests that the authors revise the manuscript according to the comments.

  1. In page 6, line 205-208, the authors need to state how much percentage/how many times has/have changed in the results, which makes the data solid.

  1. Since the authors used the abbreviations inappropriately through the manuscript, they should revise the inappropriate use. The abbreviations with the full names should be used when the corresponding terms first appeared in the text. For instance, “nitric oxide” appeared in page 2, line 32 for the first time, not in page 2, line 39. Also, “prostaglandins” in page 2, line 33, not in page 2, line 39. Furthermore, there are several “nitric oxide (NO)” and “reactive oxygen species (ROS)” in the text. Once the authors use an abbreviation in the first time, they can use it after that.

  1. In page 3, line 94, “(0.01-3 μM)” should be “(0.1-1 μM)”.

  1. In page 3, line 95, “after” should be “After”.

  1. There are two “after incubation” in page 4, line 103-104.

  1. In page 8, line 260, “f” should be “of”.

  1. There is no citation of reference after Reddy and Swamy (2015) in page 11, line 370.

  1. Indicate the wavelength in page 12, line 442.

  1. The references should be cited with numbers instead of (Adams et al., 1995) and (Raschke et al., 1978) in page 2, line 45-46 and page 12, line 442, respectively.

  1. About reference 42, “J. Immunol.” and “2007” should be in italic and bold face, respectively.

Author Response

Reviewer 2:

The authors analyzed the anti-inflammatory activity of C18-5HT, and found that C18-5HT inhibits the leukocyte migration into the SAP, the production of NO and cytokines, and the protein extravasation. The study is interesting and potentially important for the development of new anti-inflammatory drugs. To step up the quality of the manuscript, the reviewer suggests that the authors revise the manuscript according to the comments.

We really appreciate the comment.

-In page 6, line 205-208, the authors need to state how much percentage/how many times has/have changed in the results, which makes the data solid.

We included media/sd to each value. Please see page6 and 7, lines 204 to 219.

-Since the authors used the abbreviations inappropriately through the manuscript, they should revise the inappropriate use. The abbreviations with the full names should be used when the corresponding terms first appeared in the text. For instance, “nitric oxide” appeared in page 2, line 32 for the first time, not in page 2, line 39. Also, “prostaglandins” in page 2, line 33, not in page 2, line 39. Furthermore, there are several “nitric oxide (NO)” and “reactive oxygen species (ROS)” in the text. Once the authors use an abbreviation in the first time, they can use it after that.

We apologize for our mistakes. We did a revision along the text and correct the errors.

-In page 3, line 94, “(0.01-3 μM)” should be “(0.1-1 μM)”.

We corrected the error

-In page 3, line 95, “after” should be “After”.

We corrected the error

-There are two “after incubation” in page 4, line 103-104.

We corrected the error

-In page 8, line 260, “f” should be “of”.

We corrected the error

-There is no citation of reference after Reddy and Swamy (2015) in page 11, line 370.

We added this reference

-Indicate the wavelength in page 12, line 442.

We corrected the error

-The references should be cited with numbers instead of (Adams et al., 1995) and (Raschke et al., 1978) in page 2, line 45-46 and page 12, line 442, respectively.

We corrected the errors

-About reference 42, “J. Immunol.” and “2007” should be in italic and bold face, respectively.

We corrected the error